# Comparing Measurements of Vascular Diameter Using Adaptative Optics Imaging and Conventional Fundus Imaging

**DOI:** 10.3390/diagnostics12030705

**Published:** 2022-03-13

**Authors:** Thibaud Mautuit, Rachel Semecas, Stephen Hogg, Vincent Daien, Olivier Gavard, Nicolas Chateau, Tom MacGillivray, Emanuele Trucco, Christophe Chiquet

**Affiliations:** 1Department of Ophthalmology, University Hospital of Grenoble-Alpes, University Grenoble Alpes, Inserm, CHU Grenoble Alpes, HP2, 38000 Grenoble, France; t.mautuit21@orange.fr (T.M.); rachel.semecas@gmail.com (R.S.); oliv.gavard@orange.fr (O.G.); 2VAMPIRE Project, Computing (SSE), University of Dundee, Dundee DD1 4HN, UK; s.c.z.hogg@dundee.ac.uk (S.H.); e.trucco@dundee.ac.uk (E.T.); 3Department of Ophthalmology, University Hospital of Montpellier, Montpellier University, INSERM U1061, 34000 Montpellier, France; vincent.daien@gmail.com; 4Imagine Eyes, 18 rue Charles de Gaulle, 91400 Orsay, France; nchateau@imagine-eyes.com; 5VAMPIRE Project, Centre for Clinical Brain Sciences, University of Edinburgh, Edinburgh EH8 9YL, UK; t.j.MacGillivray@ed.ac.uk

**Keywords:** retinal vessel, adaptive optics, vessel diameter, IVAN software, VAMPIRE annotation tool

## Abstract

The aim of this prospective study was to compare retinal vascular diameter measurements taken from standard fundus images and adaptive optics (AO) images. We analysed retinal images of twenty healthy subjects with 45-degree funduscopic colour photographs (CR-2 Canon fundus camera, Canon™) and adaptive optics (AO) fundus images (rtx1 camera, Imagine Eyes^®^). Diameters were measured using three software applications: the VAMPIRE (Vessel Assessment and Measurement Platform for Images of the REtina) annotation tool, IVAN (Interactive Vessel ANalyzer) for funduscopic colour photographs, and AO_Detect_Artery™ for AO images. For the arterial diameters, the mean difference between AO_Detect_Artery™ and IVAN was 9.1 µm (−27.4 to 9.2 µm, *p* = 0.005) and the measurements were significantly correlated (r = 0.79). The mean difference between AO_Detect_Artery™ and VAMPIRE annotation tool was 3.8 µm (−34.4 to 26.8 µm, *p* = 0.16) and the measurements were poorly correlated (r = 0.12). For the venous diameters, the mean difference between the AO_Detect_Artery™ and IVAN was 3.9 µm (−40.9 to 41.9 µm, *p* = 0.35) and the measurements were highly correlated (r = 0.83). The mean difference between the AO_Detect_Artery™ and VAMPIRE annotation tool was 0.4 µm (−17.44 to 25.3 µm, *p* = 0.91) and the correlations were moderate (r = 0.41). We found that the VAMPIRE annotation tool, an entirely manual software, is accurate for the measurement of arterial and venular diameters, but the correlation with AO measurements is poor. On the contrary, IVAN, a semi-automatic software tool, presents slightly greater differences with AO imaging, but the correlation is stronger. Data from arteries should be considered with caution, since IVAN seems to significantly under-estimate arterial diameters.

## 1. Introduction

The retinal vasculature provides a unique window to assess non-invasively and directly vasculature segments in vivo and reflects the efficiency of the circulation and the distribution of shear stress. Over the last decade, several population-based studies have reported associations of summative measurements of the width of retinal vessels, as well as tortuosity and fractal dimension, with a wide range of subclinical (e.g., atherosclerosis, inflammation and endothelial dysfunction) [1,2] and clinical cardiovascular or ocular diseases (hypertension, diabetes mellitus, stroke, kidney and heart diseases, anterior ischemic optic neuropathy and glaucoma) [3,4,5,6].

Measurements of retinal vessel diameters are often obtained from standard fundus images by computer analysis. Several software applications have been developed, including IVAN (Interactive Vessel ANalyzer) [7], SIVA (Singapore I Vessel Assessment) [8] and VAMPIRE [9,10,11] (Vessel Assessment and Measurement Platform for Images of the Retina, Universities of Edinburgh and Dundee, UK). The VAMPIRE annotation tool supports an entirely manual measurement of retinal vessels [12]. A semi-automatic version of VAMPIRE has been used in several studies [13,14] but is not considered here. The IVAN is a semi-automatic package computing summative width measures of arteries and veins around the optic disc [15,16]. In contrast to the VAMPIRE annotation tool, the edges of the vessels are automatically located by the software.

Adaptive optics (AO) imaging systems allow improved transverse resolution compared with conventional retinal imaging by measuring the ocular wavefront aberrations and compensating for them in real time with active optical elements [17]. Detailed analysis of retinal vessels (lumen diameter of veins and arteries, and arterial parietal thickness) can be performed in vivo [18] because the lateral resolution is increased to about 2 μm. The measure of diameters by AO detect artery^®^ is also achieved automatically. Given the very high resolution of AO imaging, we assume a higher accuracy than that achievable with standard fundus camera. It is thus of particular interest to compare retinal diameters measured with high-resolution AO images with those measured from fundus pictures.

The first aim of this study was to compare vascular diameter measurements from standard fundus images and AO images, considering AO measurements as a gold standard. The second aim was to compare three different software tools.

## 2. Materials and Methods

**Subjects.** Twenty healthy subjects were included (median age: 31, range: 20–75 years old). The exclusion criteria were: pregnant or lactating women, individuals aged less than 18 years, adults under guardianship or unable to provide consent, patients with ametropia >3 diopters (D) (spherical equivalent) or with any ophthalmologic pathology. Each subject was asked for his or her medical history, especially cardiovascular risk factors (high blood pressure, diabetes, dyslipidemia, smoking). We also recorded their age, sex, weight, height and body mass index (BMI). We measured visual acuity (LogMAR), axial length (mm, IOL master™, Carl Zeiss Meditec, Iena, Germany), intraocular pressure (IOP, Tonoref 2™, NIDEK SA, Créteil, France) and their objective refraction (in diopters, Tonoref 2™, NIDEK SA, Créteil, France). After 10 min of rest, blood pressure was measured in the sitting position using an arm cuff (Dinamap Carescape V100™, GE Healthcare, Chalfont St Giles, UK). Dilation of the measured eye was obtained after instillation of one-drop tropicamide 0.5% (Mydriaticum; Thea, Clermont-Ferrand, France). First, we acquired a fundus image using a CR-2 CANON fundus camera (Canon™ Europa, Amstelveen, The Netherlands) and then an AO image using a commercial AO retinal camera (rtx1; Imagine Eyes™, Orsay, France).

The images collected were part of a prospective study using AO at the University Hospital of Grenoble (IRB# 5921). This study was conducted in accordance with the ethical principles of the Declaration of Helsinki. Subjects were included after providing written and oral informed consent.

**Image Acquisition.** A 45-degree 5184 × 3456 funduscopic colour photograph (“fundus image” in the manuscript) of the right eye centered on the optic nerve was acquired for each subject with a non-mydriatic CR-2 CANON fundus camera. In the same session, an AO image was centered on a section of the temporal superior and temporal inferior arteries and veins, at one papillary diameter from the optic nerve, in zone B. *En face* AO fundus images were obtained using a rtx1 AO retinal camera. This camera measures aberrations with a 750 nm super-luminescent diode source and corrects the wave front with an AO system operating in a closed loop. A 4° × 4° fundus area (i.e., approximately 1.2 × 1.2 mm in emmetropic eyes) is illuminated at 840 nm by a temporally low coherent light-emitting diode flashed flood source, and a stack of 40 fundus images is acquired in 4 s (10 images/s) by a charge-coupled device camera. These 40 images are averaged to obtain a single 1500 × 1500 image of 4° × 4° [19,20]. Following an alignment procedure on the cornea apex, a high-resolution image of the retina was displayed by the rtx1 to observe the chosen vessels before the acquisition. The fixation target of the rtx1 does not allow observation of the optic disc (OD) or of the vessels close to the OD. Therefore, an external light target was used for the fixation of the contralateral eye.

**Software Analysis.** The VAMPIRE annotation tool was used to obtain measurements of the vessel diameter at the same locations as those performed using the AO Artery software, in zone B. Using IVAN, a standardized grid was manually overlaid and centered on the OD. Vessels in zone B were traced automatically. In order to make measurements on the same location defined on AO image, the vessels traced automatically were truncated distally and proximally; the length of the remaining vessel segment was approximately 15 pixels. VAMPIRE annotation tool and IVAN were only used with fundus images.

To measure the lumen vessel diameter from AO images, we used the AO_Detect_Artery™ software by Imagine Eyes. The target location on a vessel was selected and analysed semi-automatically [21,22]. The software algorithms computed local values of lumen diameter, wall thickness, wall-to-lumen ratio and wall cross-sectional area. Theses analyses are supervised: when needed, the operator could readjust the positions of the wall edges with respect to the gradient intensity profile of the vessel image.

**Conversion from pixels to microns.** For images obtained using AO, the relationship between the anatomical size (mm) and the corresponding visual angle (deg) is defined by the Littmann–Bennet formula: *R*1 = 0.01306 (*x* − 1.82), with *R*1 in mm/deg and *x* the axial length in mm. The relationship between the number of pixels and the corresponding visual angle, *R*2, is a constant linked to the architecture of the rtx1, measured as 373.87 (pixel/deg).

Therefore, the relationship between the anatomical size and the size in pixels is: *R*3 = *R*1*/R*2 = 0.01306 (*x* − 1.82)/373.87.

For images obtained using the CR-2 camera, the pixel-to-mm conversion factor was obtained by dividing the average vertical optic disc diameter (ODD) over all images by the assumed average disc diameter in microns (1800 μm), following a well-established procedure [15,23]. The conversion factor was 3.8 microns/pixels.

**Comparisons of retinal vessels diameters.** The endoluminal diameter of four vessels per eye (the two major inferior and superior temporal veins and arteries) was imaged in zone B using AO and CR-2 cameras (Figure 1). First, the same segment of the vessel was measured on an AO image using the AO_Detect_Artery™ software and then on a fundus image using the VAMPIRE annotation tool and IVAN.


**Statistical Analysis.**


Statistical analysis was performed using the Statistical Package for the Social Sciences program (SPSS 17.0 for Windows, Chicago, IL, USA). The normal distribution and the homogeneity of variances of the matched data were assessed by a Shapiro–Wilk test and Bartlett’s test. Mean comparisons were then assessed by a paired *t*-test if the previous two tests were not significant, and otherwise by a Wilcoxon test. A repeated measures ANOVA was used for multiple mean comparisons. Correlations were studied using the Pearson test. Concordance was studied using intra-class correlation (ICC) and Bland–Altman graphs. For each comparison, the presence of systematic bias was assessed using a one-sample *t*-test comparing the mean difference and zero value. Proportional bias was tested by determining whether the slope of the regression line significantly differed from zero.

## 3. Results

### 3.1. Measurements of Arterial Diameters

The average of arterial diameters was of 93.6 ± 11.6 µm for AO_Detect_Artery™, 97.4 ± 12.0 µm for VAMPIRE annotation tool and 102.7 ± 15.5 µm for IVAN. The distribution of the measurements values for each software is represented in Figure 2A.

#### 3.1.1. AO and IVAN Measurements

The measurements using the AO_Detect_Artery™ software on AO image and IVAN software on fundus image were significantly correlated (r = 0.79, Figure 3A). The ICC was 0.77 (95% confidence interval (CI), 0.6; 0.87). Agreement limits ranged from −27.4 to 9.2 μm. The mean difference between techniques was 9.1 ± 9.3 µm (*p* = 0.005), which represents the systematic bias. The Bland–Altman plot indicated a significant systematic (*p* < 0.001) and a proportional bias (*p* = 0.006), both significant. IVAN measurements may be derived from AO measurements by the regression formula IVAN = 1062 × AO + 3260. For instance, for one micron of change in AO imaging, IVAN changes are estimated as 4.3 microns.

#### 3.1.2. AO and VAMPIRE Measurements

The measurements using the AO_Detect_Artery™ software on AO image and VAMPIRE annotation tool on fundus image were not significantly correlated (r = 0.12, Figure 3B). The ICC was 0.13 (95% CI, −0.19; 0.43). Agreement limits ranged from −34.4 to 26.8 μm. The mean difference between techniques was 3.8 ± 15.6 µm (*p* = 0.16). No significant systematic or proportional bias was found.

#### 3.1.3. IVAN and VAMPIRE Measurements

The measurements using the IVAN software on fundus image and VAMPIRE annotation tool on fundus image were not significantly correlated (r = 0.07, Figure 3C). The ICC was 0.07 (95% CI, −0.25; 0.39). Agreement limits ranged from −42.3 to 31.7 μm. The mean difference between techniques was 5.2 ± 18.9 µm (*p* = 0.10). No significant systematic or proportional bias was found.

There was a significant difference between the mean measurements of the three methods (AO_Detect_Artery™ software on AO image, VAMPIRE annotation tool and IVAN on fundus image, *p* = 0.0125).

### 3.2. Measurements of Venular Diameters

The average of the venular diameters was 132.6 ± 12.3 µm for AO_Detect_Artery™, 128.6 ± 19.3 for the VAMPIRE annotation tool and 132.1 ± 20.7 for IVAN. The distribution of the venular measurements for each software is represented in Figure 2B.

#### 3.2.1. AO and IVAN Measurements

The measurements using the AO_Detect_Artery™ software on AO image and IVAN software on fundus image were significantly correlated (r = 0.83, Figure 4A). The ICC was 0.83 (95% CI, −0.7; 0.91). Agreement limits ranged from −40.9 to 41.9 μm. The mean difference between techniques was 3.9 ±10.9 µm (*p* = 0.35). The Bland–Altman plot indicated a significant systematic (*p* = 0.02) bias only.

#### 3.2.2. AO and VAMPIRE Measurements

The measurements using the AO_Detect_Artery™ software on AO image and VAMPIRE annotation tool on fundus image were significantly correlated (r = 0.41, Figure 4B). The ICC was 0.41 (95% CI, 0.12; 0.64). Agreement limits ranged from −17.44 to 25.3 μm. The mean difference between techniques was 0.4 ± 21.1 µm (*p* = 0.91). The Bland–Altman plot indicated a significant slight proportional bias only (*p* = 0.04).

#### 3.2.3. IVAN and VAMPIRE Measurements

The measurements using the IVAN software on fundus image and VAMPIRE annotation tool on fundus image were significantly correlated (r = 0.54, Figure 4C). The ICC was 0.55 (95% CI, 0.28; 0.73). Agreement limits ranged from −34 to 40.9 μm. The mean difference between techniques was 3.4 ± 19.1 µm (*p* = 0.44). No significant systematic or proportional bias was found.

There was no significant difference between mean measurements of the three methods (AO_Detect_Artery™ software on AO image, VAMPIRE annotation tool and IVAN on fundus image, *p* = 0.624).

Table 1 compares the differences between each software tool, for arteries and veins. The discrepancy is normalized by the mean diameter of the vessels, in order to take into account the size difference existing between arteries and veins. Thereby, this table underlines that the agreement between software tools is better for veins than for arteries.

## 4. Discussion

This study compared vessel width measurements from three image software applications using two fundus cameras and one AO camera. The results showed that: (1) diameter measurements using IVAN were slightly different for arteries and similar for veins compared to AO measurements; AO and fundus measurements were well correlated, and the Bland–Altman plot indicated a systematic bias; (2) diameter measurements using the VAMPIRE annotation tool were slightly different for arteries and veins from AO measurements; AO and fundus measurements were well correlated for veins but not for arteries, and the Bland–Altman plot indicated no systematic bias; (3) measurements from IVAN were not significantly different from the ones from the VAMPIRE annotation tool, were significantly correlated, and the Bland–Altman plot indicated no systematic bias.

When comparing AO images and standard fundus images for arteries, IVAN measurements were well correlated and slightly underestimated, with a mean difference of 9 microns. Data from the Bland–Altman plot indicated that the two measurements were quite similar. This was not the case for the VAMPIRE annotation tool: here, results were not correlated with AO measurements, and showed larger agreement limits but no systematic or proportional bias. Differences between measurements may be mainly related to the nature of the image and the automatic detection of vessel walls. Two factors related to the acquisition of retinal image may explain the differences between measurements from AO and standard fundus images.

The first point is the higher accuracy of lumen detection using width, especially of arteries, which has also been reported for the AO-SLO technique [24]. Thus, vessel diameter from colour photographs is defined as the diameter of the column of moving blood particles [18] and does not include the transparent plasma edge stream and the vessel wall (estimated on AO images in the present study as 24.3 ± 4.8 microns). Moreover, the vessel diameter changes during the cardiac cycle generate different width variations in AO and standard fundus images. This point is more important for arterial than venular vessels, with an increase of about 3.4% in mid to late systole and a decrease thereafter towards diastole [25].

The second point is the difference in arterial measurements between IVAN and the VAMPIRE annotation tool, illustrated, on the one hand, by the systematic and proportional bias observed in IVAN measurements when compared with AO measurements, and by the absence of bias for VAMPIRE on the other hand. These two software tools were not comparable on artery measurements, and IVAN measurements from fundus images could be extrapolated more reliably from AO images due to the systematic bias and the good correlation. IVAN measurements can be extrapolated from AO measurements by the regression formula IVAN = 1.062 × AO + 3.260. Differences between IVAN and VAMPIRE may be due to the different algorithms used for vessel segmentation and width estimation, and the different length of vessel sections used for width measurements (15 pixels for IVAN, 1 pixel for VAMPIRE annotation tool). The VAMPIRE annotation tool (manual method) is the simpler method to measure the vessel diameter, using a digital caliper, but this technique is highly dependent on the vessel width estimation and placement of calipers.

When compared with AO images for veins, IVAN and VAMPIRE generated good approximations, with slight differences and good correlations. Correlation with AO measurements was better using IVAN. Many factors may be responsible for the better concordance between AO and fundus images measurements: venules are more distinct on retinal photographs, the vascular wall of veins is thinner, the lower changes across cardiac cycle [26].

From a practical point of view, our results suggest that standard fundus images provide a valuable estimation of retinal vessel diameters in healthy subjects, more accurate for veins than for arteries. This difference may be due to the often fuzzy appearance of vessel walls in fundus images compared to AO ones. This information is important since fundus images are highly available, which is not the case for AO imaging. The highest resolution of AO images makes them a desirable reference technique when comparing morphometric measurements of the retinal vasculature. In clinical research, arterial measurements are critical in systemic hypertension and other cardiovascular disease (coronary heart disease, stroke mortality) whereas venular changes in vein diameters have been associated with systemic hypertension, hyperglycaemia, obesity and increased risk of stroke [7,27,28,29].

We acknowledge some limitations. (1) The reproducibility of measurements was not evaluated in the present study since excellent inter- and/or intra-grader reproducibility was previously reported for AO detect artery software [21] and IVAN software [30,31]. (2) VAMPIRE annotation tool is a fully manual software and conclusions drawn in this study are not applicable to the VAMPIRE semi-automatic package [10,11,12]. (3) The conversion factors used for AO images and fundus images were not based on the same calculations, i.e., the Littmann–Bennett formula [32] for AO images and the assumption of 1800 microns for ODD for fundus images. The latter technique was used here since it was adopted in IVAN. We evaluated the ODD using the Littmann–Bennett formula and found an estimate of 2018 ± 159 microns, with a conversion factor of 4.2 ± 0.2, significantly different from 1800 microns (*p* < 0.001), with a conversion factor of 3.8. Therefore, the 1800 microns technique underestimates the true conversion factor in our images and is likely to contribute in some measure to the differences encountered in the comparison between measurements. (4) Pulsatility was not recorded for arteries, from AO or standard images. Using AO imaging, the image was calculated from 40 images acquired during 4 s, which is not feasible using the rtx1. On the other hand, standard images may be recorded using an ECG-gated technique to investigate pulsatility. (5) Images were recovered from healthy subjects and these results may not be applied in other populations of subjects, for instance those with diabetic retinopathy, especially if retinal new vessels and retinal haemorrhages are present. Finally, only large temporal arteries and veins were analysed at one OD from the optic nerve in zone B.

## 5. Conclusions

In conclusion, this comparative study showed that, using IVAN and the VAMPIRE annotation tool, fundus images allowed for more accurate measurements of vein diameters than of artery diameters, when compared with a gold standard defined by measurements in AO images of the same vessels and eyes. Data from arteries should be considered with caution, since IVAN seems to significantly under-estimate arterial diameters.

## Figures and Tables

**Figure 1 diagnostics-12-00705-f001:**
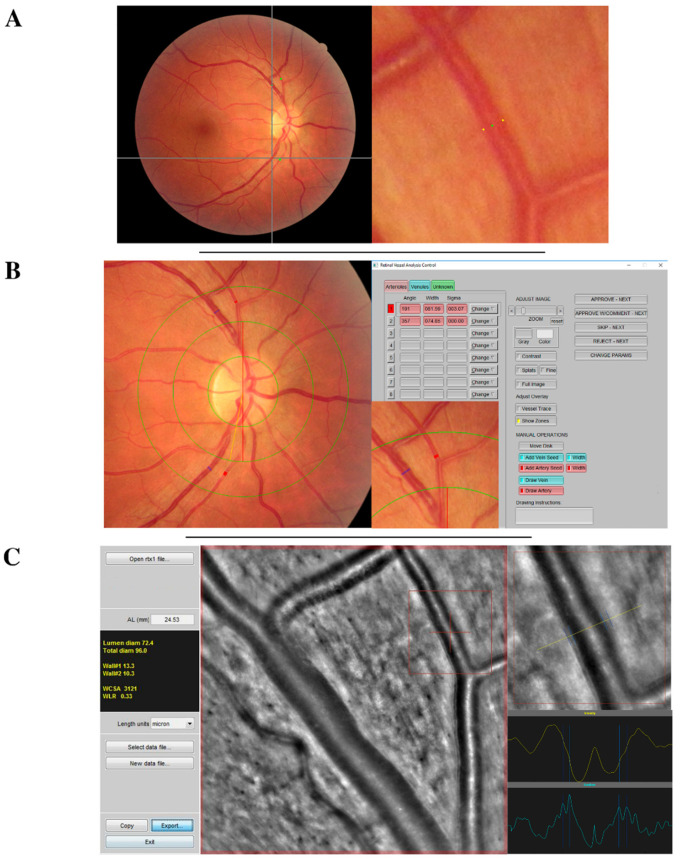
Acquisition interface of retinal vessel measurements (**A**) using vessel analysis using VAMPIRE annotation tool, (**B**) IVAN software, or (**C**) adaptive optics camera and AO_Detect_Artery™. This figure briefly presents the acquisition interface of each vessel measurement technique that has been systematically used to measure four vessels (the two temporal arteries and the two temporal veins) of each retinal photograph. Vessel measurements were performed at the same section with all software tools. In this example the measured section of the superior temporal artery is framed in red.

**Figure 2 diagnostics-12-00705-f002:**
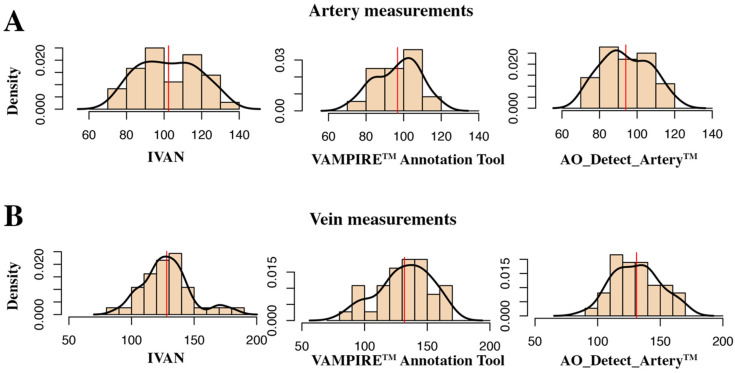
Estimated distributions of diameter values by software tool, for artery measurements (**A**) and vein measurements (**B**) and software tool.

**Figure 3 diagnostics-12-00705-f003:**
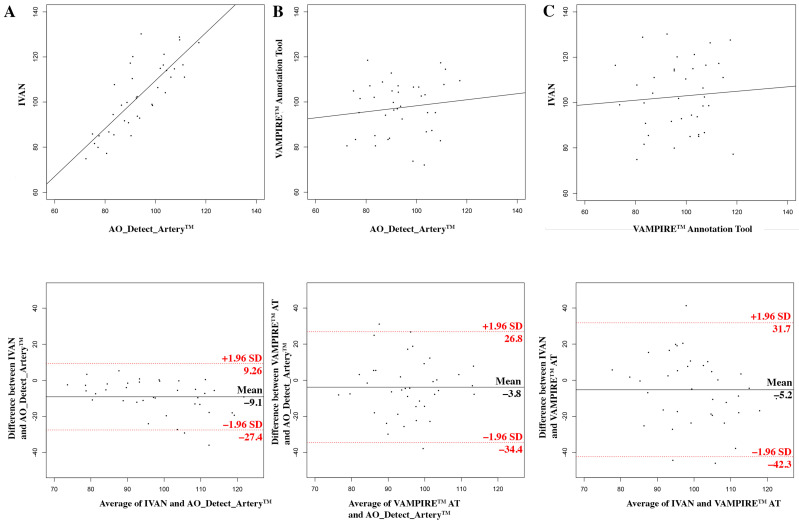
Measurements of retinal artery lumens using AO and fundus camera images. Top row: scatter plots of vessel diameter measurements with the result of the Pearson correlation tests. Bottom row: Bland–Altman plots (see axis labels for measures, where compared). The top and bottom lines visualize the 95% limits of agreement (2SD) (**A**) using the AO_Detect_Artery™ software on AO image vs. IVAN software on fundus image; (**B**) using the AO_Detect_Artery™ software on AO image vs. VAMPIRE AT (annotation tool) on fundus image; (**C**) using IVAN software vs. VAMPIRE AT on fundus image.

**Figure 4 diagnostics-12-00705-f004:**
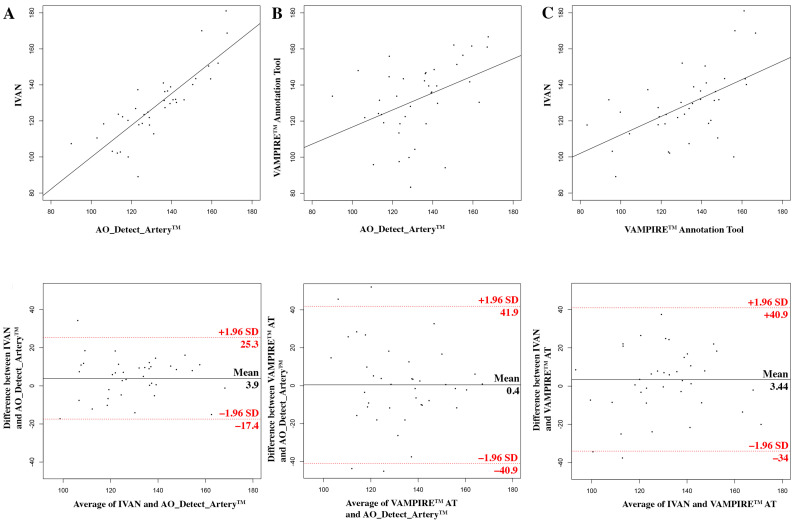
Measurements of retinal vein lumens using AO and fundus camera images. Top row: scatter plots of vessel diameter measurements with the result of the Pearson correlation tests. Bottom row: Bland–Altman plots (see axis labels for measures, where compared). The top and bottom lines visualize the 95% limits of agreement (2SD) (**A**) using the AO_Detect_Artery™ software on AO image vs. IVAN software on fundus image; (**B**) using the AO_Detect_Artery™ software on AO image vs. VAMPIRE AT (annotation tool) on fundus image; (**C**) using IVAN software vs. VAMPIRE AT on fundus image.

**Table 1 diagnostics-12-00705-t001:** Differences between software tools for arteries and for veins.

	For Arterial Diameters		For Venular Diameters	
Mean difference between	In µm	In % of the mean total diameter	In µm	In % of the mean total diameter
AO and IVAN	9.1	9.2	3.9	2.9
AO and VAMPIRE	3.8	3.8	0.4	0.3
IVAN and VAMPIRE	5.2	5.3	3.4	2.5

## Data Availability

Data may be available upon request.

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
