# Peer review of "Comparing Measurements of Vascular Diameter Using Adaptative Optics Imaging and Conventional Fundus Imaging"

_diagnostics, 2022, doi:10.3390/diagnostics12030705_

Round 1

Reviewer 1 Report

The authors describe their results about a comparison of measurements of retinal vessel diameters (arterial, venous) from images done by a standard fundus camera and an adaptic optics (AO) camera in 20 healthy subjects. They used 3 different software tools (VAMPIRE [manual], IVAN [semi-automatic] and AO-Detect-Artery. AO pictures were taken as references. Analyses by VAMPIRE were accurate, but with poor correlation, in contrast to IVAN analyses with differences to AO (underestimation of diameter of arteries), but high correlation. The probable reasons for these differences and the few limitations of the study are explained.

In summary, this manuscript is well written, interesting to read, with excellent methods and statistics, and meaningful figures. The clinical impact is high, because the analysis of retinal vessels is a surrogate for the vascular situation of the whole body.

Two comments: please explain OD at the first time (optic disc, line 112), and please inform about the mean age and range of the age of the subjects.

Author Response

Please explain OD at the first time (optic disc, line 112), and please inform about the mean age and range of the age of the subjects.

  • OD have been spelled out for the first time, page 3, line 112
  • Median age and range of age of healthy subjects was added, page 2, line 77

Reviewer 2 Report

the authors wrote an interesting paper

I'd suggest to expand the limitations of the study 

Author Response

I'd suggest to expand the limitations of the study

According to the comment, we add two limitations of the study, page 10, lines 428-432:

“(5) Images were recovered from healthy subjects and these results may not be applied in other population of subjects, for instance with diabetic retinopathy, especially if retinal new vessels and retinal hemorrhages are present. Finally, only large temporal arteries and veins were analyzed at one OD from the optic nerve in zone B.”

Reviewer 3 Report

In this manuscript, authors have reported a comparative study of fundus vessels size measured with standard digital fundus images and adaptive optics (AO) images.Diameters were measured using three software applications: VAMPIRE® Annotation Tool, IVAN for  funduscopic color photographs, and AO_Detect_Artery™ for AO images. They conclude while considering AO imaging as a reference, that the VAMPIRE Annotation Tool, is accurate for the arterial and venular diameters measurements, but  the correlation is poor. Whereas, IVAN, provided  slightly higher differences with a stronger correlation, in AO imaging. This is a small interesting piece of work. Manuscript is overall well drafted. Manuscript could be considered publishing after addressing the following minor concerns/suggestions.

  1. Can authors provide the vessel analysis with different approaches with an example such as CNVs. How much this would help to quality the growth and changes in the vasculature growth such as CNVs (PMID: 32128346)
  2. Can authors perform a similar analysis with OCT fundus images and make a comparsion between the approaches as well as with the above measurements. This will provide a comprehensive conclusion of clinical utility of your findings.
  3. The control panel shown in Figure 1B is of poor quality and pixelated. Use a high quality graphics.

Author Response

  1. Can authors provide the vessel analysis with different approaches with an example such as CNVs. How much this would help to quality the growth and changes in the vasculature growth such as CNVs (PMID: 32128346)

We understand the need of measurement of CNVs in clinical practice using quantification of vessel diameters, surface and number of vessels. However, this suggestion of work is out the scope of this project and out of the technical properties of softwares used in this study. IVAN and VAMPIRE softwares algorithms may only detect and measure retinal vessels and not subfoveal vessels. It is also true for proliferative new vessels at the surface of the retina, such in diabetic retinopathy.

  1. Can authors perform a similar analysis with OCT fundus images and make a comparison between the approaches as well as with the above measurements. This will provide a comprehensive conclusion of clinical utility of your findings.

This study has not been planned with OCT-angiography imaging, therefore it will be not possible to compare data from OCT with standard images. In addition, the goal of the study focused on the comparison between IVAN and VAMPIRE softwares with AO imaging, which can be the gold standard for the measurement of retinal vessel diameter.

  1. The control panel shown in Figure 1B is of poor quality and pixelated. Use a high quality graphics.

Figure 1 has been provided in 300 dpi, and when increasing the size of the image, the text is readable. We can provide the figure 1 in a separate file, which will be convenient for the reviewer and readers.

As requested, English editing has been reviewed by co-authors, E. Trucco and Tom McGillivray, from UK

Round 2

Reviewer 2 Report

Authors improved the paper